



# Detecting dynamical anomalies in time series from different palaeoclimate proxy archives using windowed recurrence network analysis

Jaqueline Lekscha[1,2] and Reik V. Donner[1,3]

[1]Potsdam Institute for Climate Impact Research (PIK) – Member of the Leibniz Association, 14473 Potsdam, Germany
[2]Department of Physics, Humboldt University, 12489 Berlin, Germany
[3]Department of Water, Environment, Construction and Safety, Magdeburg-Stendal University of Applied Sciences, 39114 Magdeburg, Germany

**Correspondence:** lekscha@pik-potsdam.de

**Abstract.** Analysing palaeoclimate proxy time series using windowed recurrence network analysis (wRNA) has been shown to provide valuable information on past climate variability. In turn, it has also been found that the robustness of the obtained results differs among proxies from different palaeoclimate archives. To systematically test the suitability of wRNA for studying different types of palaeoclimate proxy time series, we use the framework of forward proxy modelling. For this, we create

artificial input time series with different properties and, in a first step, compare the time series properties of the input and the model output time series. In a second step, we compare the areawise significant anomalies detected using wRNA. For proxies from tree and lake archives, we find that significant anomalies present in the input time series are sometimes missed in the input time series after the nonlinear filtering by the corresponding models. For proxies from speleothems, we observe falsely identified significant anomalies that are not present in the input time series. Finally, for proxies from ice cores, the wRNA

results show the best correspondence with those for the input data. Our results contribute to improve the interpretation of windowed recurrence network analysis results obtained from real-world palaeoclimate time series.

## 1 Introduction

Palaeoclimate proxy time series from archives such as trees, lakes, speleothems, or ice cores play an important role in past climate reconstructions (Bradley, 2015). Apart from information on climatic boundary conditions such as local and global

mean temperatures and precipitation sums, the analysis of the proxy time series using non-linear methods offers the possibility to study dynamical anomalies in past climate variability (Marwan et al., 2002, 2003; Trauth et al., 2003; Marwan et al., 2009; Donner et al., 2010b; Franke and Donner, 2017). Due to the limitations of palaeoclimate proxy time series in terms of non-uniform sampling of the data points, uncertainties in dating and measurement, the contamination with noise, and the often non-unique interpretation of the climate signal within the proxy, not all methods are suitable to gain reliable information from

all data (Goswami et al., 2018; Lekscha and Donner, 2018).

Windowed recurrence network analysis (wRNA) has already been successfully used to detect dynamical anomalies in time series from different palaeoclimate archives (Donges et al., 2011a, b; Eroglu et al., 2016; Lekscha and Donner, 2018). But it has





also been observed that this method often yields high numbers of false positive significant points, and, that not all palaeoclimate archives give equally robust results. In order to improve the interpretation of results obtained with wRNA for real-world time
series, we here systematically test the suitability of the approach for analysing palaeoclimate data from different types of archives by employing proxy system models.

Proxy system models are forward models that simulate the formation of a palaeoclimate proxy based on the systemic understanding of that proxy (Evans et al., 2013). That is, given the climate input variables, the proxy system model outputs a proxy time series. We here use intermediate complexity models for tree ring width, branched glycerol dialkyl glycerol tetraethers
in lake sediments, and the isotopic composition of speleothems and ice cores (Tolwinski-Ward et al., 2011; Dee et al., 2015, 2018). We first create artificial climate input time series with different properties. In particular, we consider two stochastic processes, Gaussian white noise and an autoregressive process of order 1, and two non-stationary time series from the paradigmatic Rössler and Lorenz systems. Additionally, we use climate input from the last millennium reanalysis project as an estimate of realistic past climate variability (Hakim et al., 2016; Tardif et al., 2019). We then compare the input and the model output with
respect to the properties of the time series and the results of wRNA. To quantify significant dynamical anomalies, we use an areawise significance test that was recently introduced for wRNA (Lekscha and Donner, in press).

With this work, we want to contribute to a better understanding and an improved interpretation of results obtained with wRNA for palaeoclimate applications. We first introduce our analysis framework in Sec. 2, the proxy system models in Sec. 3, and the input time series in Sec. 4. Then, we show and discuss the time series properties and the results of the wRNA for the
different input and model output time series in Sec. 5 before we present our main conclusions in Sec. 6. Additional information and figures are included in the Supplementary Material accompanying this paper.

## 2 Analysis framework

### 2.1 Phase space reconstruction

The first step when analysing data using recurrence based approaches is to reconstruct the higher-dimensional phase space
of the system from the measured univariate time series $x(t)$ with observations made at times $\{t_i\}_{i=1}^N$. For this, we here use uniform time delay embedding (Takens, 1980; Packard et al., 1980). Time delay embedding relates the higher-dimensional coordinates of the system's phase space to delayed versions of the measured, univariate time series

$$x(t_i) \rightarrow \boldsymbol{x}(t_i) = \{x(t_i), x(t_i - \tau), \ldots, x(t_i - (m-1)\tau)\}. \tag{1}$$

Here, $m$ denotes the embedding dimension and $\tau$ is the embedding delay. The embedding theorem of Takens guarantees
that, when choosing the embedding dimension larger than twice the box-counting dimension of the original attractor, the reconstructed and the original system's attractor are related by a smooth one-to-one coordinate transformation with smooth inverse, independent of the choice of the delay (Takens, 1980). That is, analysing the dynamics in the reconstructed phase space can give information on the original system dynamics.



To determine the embedding parameters in practical applications, the method of false nearest neighbours (Kennel et al.,
1992) can be used for estimating the embedding dimension, while the first zero of the autocorrelation function or the first
minimum of the mutual information can serve as an estimate for the embedding delay (Abarbanel et al., 1993; Fraser and
Swinney, 1986; Kantz and Schreiber, 2004). We here use embedding dimension $m = 3$ as a compromise between the number
of available data points and the minimum number of independent coordinates as indicated by the false nearest neighbour
criterion. For estimating the embedding delay, we use the first zero of the autocorrelation function.

## 2.2 Windowed recurrence network analysis

We analyse the embedded time series $\boldsymbol{x}(t)$ by means of windowed recurrence network analysis (wRNA), using a sliding
window approach and dividing the embedded time series into windows of width $W$ with mutual offset $dW$. For each of those
$N' = (N - (m-1)\tau - W)/dW$ windows, we construct a network from the time series by identifying the different $\boldsymbol{x}_i = \boldsymbol{x}(t_i)$
as nodes of the network and drawing a link between two nodes $\boldsymbol{x}_i$ and $\boldsymbol{x}_j$ if they are mutually closer in phase space than some
threshold $\epsilon$. For this purpose, we measure the distances in phase space using the maximum norm

$$\|\boldsymbol{x}\|_\infty = \max_{k=1,\dots,m} \left\{ x^{(k)} \right\}. \tag{2}$$

Then, the resulting recurrence network can be described by its adjacency matrix $\mathbf{A}$ with entries

$$A_{i,j}(\epsilon) = \theta \left( \epsilon - \|\boldsymbol{x}_i - \boldsymbol{x}_j\| \right) - \delta_{i,j}. \tag{3}$$

where $\theta(\cdot)$ is the Heaviside function and the delta function $\delta_{i,j}$ excludes self-loops in the network. Here, we select the threshold
adaptively by choosing a fixed edge density $\rho = 0.05$. This means, we choose the threshold $\epsilon$ such that a fraction $\rho$ of all possible
links in the network are realised.

From the adjacency matrix, we can estimate various network properties. In the course of this work, we will restrict ourselves
to the network transitivity

$$\mathcal{T} = \frac{\sum_{v,i,j} A_{v,i} A_{i,j} A_{j,v}}{\sum_{v,i,j} A_{v,i} A_{j,v}}, \tag{4}$$

which gives the probability that two randomly chosen neighbours of a randomly chosen node are mutually connected. Network
transitivity is particularly well suited for detecting dynamical anomalies in non-stationary time series, since this network
measure has been shown to be closely related to the dimensionality of the system dynamics (Donner et al., 2010a, 2011b).
Specifically, the quantity $\log \mathcal{T}/\log(3/4)$ corresponds to a generalised fractal dimension (Donner et al., 2011a). Thus, low
values of the network transitivity can be related to higher-dimensional dynamics and vice versa.

For the analysis performed here, we repeat the wRNA for different values of the window width $W \in [100, 300]$ with step size
$\Delta W = 1$ in order to check the robustness of the results for this analysis parameter. Thus, the resulting values of the network
transitivity can be stored in some matrix $\mathbf{Q}$ which is of size $N_W \times N'$ with $N_W$ being the number of window widths for which
the analysis has been performed and $N'$ being the number of windows into which the original time series is divided. We choose
the offset of the windowed analysis to be always $dW = 1$.



### 2.3 Significance testing and confidence levels

In order to identify dynamical anomalies from the resulting network transitivity, we first perform a pointwise significance test using random shuffling surrogates. That is, for every window width $W$, we randomly draw $N_s = 1,000$ times $W$ embedded state vectors from $\boldsymbol{x}(t)$ and calculate the network transitivity of this set of $W$ state vectors. We then take the empirical 2.5th and 97.5th percentiles from the $N_s$ realisations to obtain a two-sided confidence interval of $s_{pw} = 95\%$. All transitivity values of the original analysis results that fall outside this interval are considered to show pointwise significant anomalies, that is, the null hypothesis of the corresponding values of transitivity resulting from random data with the same amplitude distribution as the original data is rejected. The result of this pointwise significance test can be summarised in the binary significance matrix $\mathbf{S}^{pw}$ of the same size as the matrix of the transitivity results $\mathbf{Q}$. The entries of $\mathbf{S}^{pw}$ equal one if the corresponding value of the transitivity has been found to be pointwise significant, and zero otherwise.

As intrinsic correlations of the analysis results in both, the time domain (due to the short offset $dW$) and the window width domain can lead to patches of false positives in the pointwise significance matrix, we additionally perform an areawise significance test (Lekscha and Donner, in press; Maraun et al., 2007). A pointwise significant point is defined to be areawise significant, if all neighbouring points within a given rectangle are also pointwise significant, i. e., if the pointwise significant point lies within a patch of pointwise significant points that is larger than this rectangle. The side lengths of the rectangle depend on the intrinsic correlations of a chosen null model that are estimated as detailed in the following.

First, we choose a null model for which we want to perform the areawise test. We here employ a data-adaptive null model using iterative amplitude-adjusted Fourier transform surrogates of the original time series (Schreiber and Schmitz, 2000). That is, we test whether the same analysis results could have been obtained from data with the same amplitude distribution and linear correlation structure as the original data but that are otherwise random, i. e., originate from linear stochastic processes with prescribed correlations. We thus create $N_{da} = 100$ such surrogate data sets and perform the same analysis on them as on the original data. We then quantify the intrinsic correlations by calculating correlation functions in the time and the window width domain for the different values of the window width $W$. The corresponding preliminary decorrelation lengths are obtained as the values of the time lag or window width where the correlation function falls below a threshold that we set to $1/e$ for the time and $2/e$ for the window width domain. Those different values of the threshold can be explained by the different behaviour of the correlation function in the different domains and are set in accordance with the findings in Lekscha and Donner (in press). We then estimate the final decorrelation lengths $l_t$ and $l_W$ of the two domains by applying a linear fit to the preliminary decorrelation length as a function of the window width

$$l_{t/W} = m_{t/W}W + n_{t/W}. \tag{5}$$

Finally, we define the areawise significance matrix $\mathbf{S}^{aw}$ to have entries $S_{i,j}^{aw} = 1$ if the relation

$$\sum_{p=i-(l_t/2-1)}^{i+(l_t/2-1)} \sum_{q=j-(l_W/2-1)}^{j+(l_W/2-1)} S_{p,q}^{pw} = (l_t-1)(l_W-1) \tag{6}$$





holds, and to have entries $S^{aw}_{i,j} = 0$ otherwise. A detailed and more general description of the areawise significance test can be found in Lekscha and Donner (in press).

## 3 Proxy system models

Forward modelling of palaeoclimate proxies offers the possibility to gain insights into the underlying processes that influence
the sensitivity of a given proxy to climate variations and can thus be used to investigate characteristic properties of time series of different palaeoclimate archives and their implications for further analyses. We here use four models for typical proxies from tree rings, lake sediments, ice cores and speleothems, respectively, in order to test how well dynamical anomalies can be identified when applying wRNA to time series originating from those archives.

Generally, a proxy system model can be divided into an environment, a sensor, an archive and an observation sub-model (Evans
et al., 2013). The environment model can be used to model the environmental factors that the archive is sensitive to using the climatic input variables. The sensor model then describes how the archive reacts to the environment, and the archive model accounts for how this reaction is written into the archive. Finally, an observation model can be used to simulate uncertainties in dating or measurements. Here, depending on the archive, we use different combinations of the environment, sensor and archive sub-models, while neglecting possible dating and measurement uncertainties. In the following, we will give a brief overview
over the model structures and parameters. A full description of the models can be found in the corresponding references.

### 3.1 Tree rings

Tree rings are one of the most important archives for palaeoclimate reconstructions of the last millennium (Bradley, 2015; St. George, 2014; St. George and Esper, 2019). To model the tree ring width as a function of time at a particular site, we use the intermediate complexity model Vaganov-Shashkin-Lite (VS-Lite) (Tolwinski-Ward et al., 2011). This is a reduced
version of the full Vaganov-Shashkin model (Vaganov et al., 2006) and requires monthly input data of temperature $T$ and either precipitation $P$ or soil moisture $M$. Additional model parameters are thresholds for temperature and soil moisture below which growth is not possible and above which growth is optimal $(T_1, M_1, T_2, M_2)$, the latitude of the site $\Phi$, and integration start and end months $I_0$ and $I_f$ that set the period over which climate is responsible for growth in a given year.

If precipitation is given, the Leaky Bucket model (Huang et al., 1996) is used as environment model to calculate the soil
moisture $M(t)$ based on the water balance in soil. This model requires additional parameters such as the initial moisture content of the soil $M_0$, minimum and maximum soil moisture content $M_{\mathrm{min,max}}$, runoff parameters $m$, $\alpha$ and $\mu$ and the root depth $d_r$. The sensor model for the tree ring width then basically relies on the principle of limiting factors (Fritts, 1976), that is, it assumes that tree ring growth is limited by the resource that is the scarcest for optimal growing conditions, i. e., in this





case, the temperature or the soil moisture. A temperature-based growth response is calculated as

$$
\quad g_T(t) = \begin{cases} 0 & \text{if } T(t) < T_1 \\ \dfrac{T(t) - T_1}{T_2 - T_1} & \text{if } T_1 \leq T(t) \leq T_2 \\ 1 & \text{if } T_2 > T(t) \end{cases} \tag{7}
$$

and similarly, a growth response $g_M(t)$ is calculated for soil moisture. In addition, a third insolation-based growth response $g_E(t)$ is calculated based on the mean of the normalised lengths of the day for each month. The total growth response $g(t)$ of the tree to the climatic input is then given as the minimum of the temperature and moisture based growth responses modulated by the insolation based growth response,

$$
\quad g(t) = g_E(t) \min \{g_T(t), g_M(t)\}. \tag{8}
$$

To obtain the annual growth response from those monthly data, the model integrates the growth response $g(t)$ over those months that are specified as the start and end months $I_0$ and $I_f$. Finally, the annually resolved time series of tree ring width anomalies is obtained by normalising the annual growth response to zero mean and unit variance.

To set the model parameters to realistic values, we use an exemplary real-world data set of a local tree ring width index chronology from eastern Canada ($54.2°$ N, $70.3°$ W) that was previously used for regional summer temperature reconstruction (Gennaretti et al., 2014). The quality of this data set with respect to data homogeneity, sample replication, growth coherence, chronology development, and climate signal has been ranked high (Esper et al., 2016). Regional average annual temperature is about $-3.8°$C with average maximum temperatures of $16°$C in July and minimum temperatures of $-23°$C in January. The average annual precipitation sum is $693$ mm with a monthly minimum of about $30$ mm in February/March and a maximum of about $100$ mm in September. This information has been used in order to choose the mean and standard deviations of the climatic input variables and their annual cycles. An overview of the climate input and also the model parameters is given in Table 1. To determine the threshold parameters, we used the Bayesian parameter estimation as suggested in Tolwinski-Ward et al. (2013). The parameters for the Leaky Bucket model are chosen as recommended in Tolwinski-Ward et al. (2011).

### 3.2 Lake sediments

Records from lake sediments are available from many regions worldwide and can provide information about past temperatures and precipitation, depending on the regional boundary conditions and measured proxy (Cohen, 2003). We here model branched glycerol dialkyl glycerol tetraethers (brGDGTs) from lacustrine archives that have been related to air temperatures by using one of the sensor models provided in the PRYSM v2.0 framework (Dee et al., 2018). For this, as model input, a time series of mean annual air temperature $T$ is required.

BrGDGTs are produced by bacteria and their degree of cyclisation and methylation has been related to soil temperatures, lake pH, and also to mean annual air temperatures (Weijers et al., 2007; De Jonge et al., 2014; Russell et al., 2018). The degree of methylation is quantified by a methylation of branched tetraether (MBT) index. The sensor model employs the $\text{MBT}'_{5\text{ME}}$





**Table 1.** Climatic input variables and model parameters for the tree ring width model as derived from the eastern Canada data (Gennaretti et al., 2014).

| variable | description | value |
|---|---|---|
| $T_m$ | mean temperature at site | $-3.8°$ C |
| $P_m$ | mean annual precipitation sum | $693\,\mathrm{mm}$ |
| $\Phi$ | latitude of site | $54.2°\mathrm{N}$ |
| $I_{0,f}$ | integration period influencing growth | $[1, 12]$ |
| $T_{1,2}$ | temperature thresholds for growth | $[5.8, 17]°\mathrm{C}$ |
| $M_{1,2}$ | soil moisture thresholds for growth | $[0.032, 0.24]$ |
| $M_{\mathrm{min,max}}$ | minimum/maximum soil moisture content | $[0.01, 0.76]$ |
| $M_0$ | soil moisture content at start of simulation | $0.2$ |
| $m$ | runoff parameter | $4.886$ |
| $\alpha$ | runoff parameter | $0.093\,\mathrm{month}^{-1}$ |
| $\mu$ | runoff parameter | $5.8$ |
| $d_r$ | root depth | $1000\,\mathrm{mm}$ |

index that only uses 5-methyl isomers. In particular, the calibration of Russell et al. (2018) is used, in which the mean annual air temperature is related to the $\mathrm{MBT}'_{5\mathrm{ME}}$ index via the equation

$$MBT'_{5\mathrm{ME}} = (T + 1.21)/32.42. \tag{9}$$

The archive model then accounts for bioturbation and mixing of the sediments using the TURBO2 model (Trauth, 2013). TURBO2 models the benthic mixing effects on individual sediment particles by assuming that in a mixed layer of a specified thickness on top of a sediment core, instantaneous mixing of the sediment particles occurs, while the rest of the core is not affected by the mixing. In addition to the time series of the sensor model output, the archive model requires three further input

parameters, the abundance of the signal carrier over time (abu), the mixed layer thickness over time (mxl), and the number of fossil foraminifera on which the proxy signal is measured (numb). The model then returns time series of original and bioturbated abundances and corresponding proxy signatures for the original and a second virtual species that is required to keep the total abundance of all species constant over time. The bioturbated proxy signatures of the first species are then used as final proxy for the mean annual air temperature.

To tune the climatic input variables, we use the climatic setup corresponding to the one used for the tree ring archive in eastern Canada, while for the model parameters, we use typical values for lake sediments that are taken as default in the PRYSM implementation of the lake archive model (Dee et al., 2018). In particular, it should be noted that for the abundances of the input species and the mixed layer thickness, we use constant values over time. The climatic and model input parameters are specified in Table 2.





**Table 2.** Climatic input variables and model parameters for the lake sediment model as derived from the eastern Canada data (Gennaretti et al., 2014).

| variable | description | value |
|---|---|---|
| $T_m$ | mean temperature at site | $-3.8°$ C |
| abu | abundances of input species | 200 |
| mxl | mixed layer thickness | 4 |
| numb | amount of measured foraminifera | 10 |

### 3.3 Speleothems

Oxygen isotope fractions of speleothems have been shown to provide valuable insights into past climate variability (Wong and Breecker, 2015). We here model the isotopic composition of speleothems by using the speleothem model presented within the PRYSM framework (Dee et al., 2015). This intermediate-complexity model is based on the model in Partin et al. (2013) and requires the mean annual temperature $T$ and the mean of the precipitation-weighted annual isotopic composition of the precipitation $\delta^{18}O_P$ as input. Additionally, the ground water residence time $\tau_{\mathrm{gw}}$ has to be specified.

The sensor model covers processes in the karst and the cave, while processes in the soil such as evapotranspiration are neglected. The model filters the $\delta^{18}O_P$ signal by applying an aquifer recharge model to simulate the storage and thus the mixing of water of different ages in the karst. This process is parameterised by the mean transit time $\tau_{\mathrm{gw}}$. The isotopic composition of the cave water is then given as the convolution ($*$) of $\delta^{18}O_w$ with the impulse response of the aquifer recharge model $g(t) = \tau_{\mathrm{gw}}^{-1} \exp(-t\tau_{\mathrm{gw}})$ for $t > 0$:

$$\delta^{18}O_d = g(t) * \delta^{18}O_P. \tag{10}$$

Finally, to obtain the isotopic composition of the flowstone calcite $\delta^{18}O_c$, the model implements a temperature-dependent fractionation (Wackerbarth et al., 2010)

$$\delta^{18}O_c = \frac{\delta^{18}O_d + 1000}{1.03086} \exp\left(\frac{2780}{T_a^2} - \frac{2.89}{1000}\right) - 1000 \tag{11}$$

with the temperature $T_a$ being the decadal average of $T$ that is calculated using a Butterworth filter (Zumbahlen, 2008).

The parameters for the speleothem $\delta^{18}O$ model are tuned by using the data of stalagmite DA from Dongge cave (Wang et al., 2005) which is one of the most studied speleothem data sets. Dongge cave is located in Southern China ($25.3°$ N, $108.1°$ E) at an elevation of $680\,$m and the data could be related to the history of the Asian monsoon of the last $9,000$ years. The proxy values have an average of $-8.05\,‰$ and the average temperature in the cave is $15.6°$C. The mean of the precipitation-weighted isotopic composition of the input is chosen such that the average of the model output data equals the average of the Dongge cave proxy data. There is no information available about the mean transit time, i. e., the time the water has spent inside the karst before entering the cave. We here choose to use an average transit time of five years, which is slightly larger than the average sampling rate of the data (4.2 years). An overview of the input variables and model parameters is given in Table 3.


**Table 3.** Climatic input variables and model parameters for the speleothem $\delta^{18}$O model as derived from the Dongge cave data (Wang et al., 2005).

| variable | description | value |
|---|---|---|
| $T_m$ | mean temperature at site | $15.6^\circ$ C |
| $\delta^{18}$O$_P$ | mean isotopic composition of precipitation | $-8.05\,‰$ |
| $\tau_{\mathrm{gw}}$ | mean aquifer transit time | 5 years |

### 3.4 Ice cores

Proxy time series from ice cores have been used in a variety of contexts to study past climate variability on different time scales (Jouzel, 2013; Thompson et al., 2005). As for the speleothem $\delta^{18}$O, we use the model for ice core $\delta^{18}$O that is implemented and presented within the PRYSM framework (Dee et al., 2015). The model requires the precipitation-weighted mean annual isotopic composition of the precipitation $\delta^{18}$O$_P$ as input. Additional parameters are the mean temperature at the site $T_m$, the altitude of the glacier $z$, the mean surface pressure $p$, the mean accumulation rate at the site $A$, and the total depth of

the core $h_{\mathrm{max}}$ which is given by the time span of the observations times the average accumulation rate.

The sensor model corrects the isotopic composition of the precipitation for the altitude of the glacier by using the relation

$$\delta^{18}\mathrm{O}_{\mathrm{ice}} = \delta^{18}\mathrm{O}_P + \frac{z}{100}a, \tag{12}$$

with $a$ describing the altitude effect. The archive model then accounts for compaction and diffusion within the ice core. First, the density of the core has to be calculated as a function of the depth of the core. For this, an adapted version of the firn

densification model by Herron and Langway is used (Herron and Langway, 1980). From the density and the mean temperature $T_m$, we can then compute the diffusion length $\sigma$ within the core as a function of the depth $h$ (Johnsen et al., 2000). This, in turn, is used to calculate the final proxy time series $\delta^{18}$O$_d$ by convolving the isotopic signal of the ice $\delta^{18}$O$_{\mathrm{ice}}$ with a Gaussian kernel function with standard deviation equalling the diffusion length at a given depth,

$$\delta^{18}\mathrm{O}_d = \frac{1}{\sigma\sqrt{2\pi}}\exp\left(\frac{-h^2}{2\sigma^2}\right) * \delta^{18}\mathrm{O}_{\mathrm{ice}}, \tag{13}$$

where the convolution is again denoted by the asterisk ($*$).

To tune the climatic input variables and model parameters of the ice core $\delta^{18}$O model, we use an exemplary real world data set of the Quelccaya ice cap (Thompson et al., 2013) which is one of the most studied ice core data sets outside the polar regions. The Quelccaya ice cap is located in the Peruvian Andes ($13^\circ56'$ S, $70^\circ50'$ W) at an altitude of $5670$ m above sea level. The average accumulation rate is $1.15$ m water equivalent per year and the mean $\delta^{18}$O$_{\mathrm{ice}}$ is $-17.9\,‰$. The average annual

temperature over the last decade at the Quelccaya ice cap is given as $T_m = -3.99^\circ$ C (Yarleque et al., 2018). An overview of the climatic input variables and the model parameters can be found in Table 4.





**Table 4.** Climatic input variables and model parameters for the ice core $\delta^{18}$O model as derived from the Quelccaya ice cap data (Thompson et al., 2013; Yarleque et al., 2018).

| variable | description | value |
|---|---|---|
| $\delta^{18}\text{O}_P$ | mean isotopic composition of precipitation | $-3.75\,‰$ |
| $T_m$ | mean temperature at site | $-3.99°\,\text{C}$ |
| $A$ | average accumulation rate at site | $1.15\,\text{m w.e./a}$ |
| $p$ | mean surface pressure at site | $1\,\text{Atm}$ |
| $z$ | altitude of site | $5670\,\text{m}$ |
| $\rho_0$ | surface density of snow at site | $300\,\text{kg/m}^3$ |
| $a$ | altitude effect | $-0.25\,‰/100\,\text{m}$ |

## 4 Input data

We now introduce the data sets that we use as input for the proxy system models. We first consider two stationary stochastic processes, namely Gaussian white noise (GWN) and an autoregressive process of order 1 (AR(1) process) to evaluate whether such input can lead to the detection of dynamical anomalies in the proxy time series. Then, we consider non-stationary versions of the two well-known Rössler (ROS) and Lorenz (LOR) systems. For all those processes, time series of length $N = 1,000$ are independently created to describe temperature, precipitation and precipitation-weighted oxygen isotope fractions as detailed below, where the precipitation is proportional to negative temperature. Additionally, we use data from the last millennium reanalysis project (Hakim et al., 2016; Tardif et al., 2019). Then, as for the tree ring width model monthly input is required, an annual cycle is added to the temperature and precipitation data. The amplitude of the annual cycle is chosen according to the climatic boundary conditions of the tree ring width index time series from eastern Canada presented in Sec. 3. Finally, yearly means for temperature and sums for precipitation are calculated for those models that require yearly input. The input time series are normalised to zero mean and unit standard deviation, and for each model, the mean is adjusted to the corresponding climatic boundary conditions.

### 4.1 Gaussian white noise

For the case of GWN, we draw $N$ data points independently at random from the probability distribution

$$p_G(x) = \frac{1}{\sqrt{2\pi}\sigma} \exp\left(\frac{(x-\mu)^2}{2\sigma^2}\right), \tag{14}$$

where $\sigma = 1$ is the standard deviation and $\mu = 0$ is the mean of the distribution. We do not expect to detect significant dynamical anomalies from time series created in this way as the process is stationary.





## 4.2 Autoregressive process of order 1

For the AR(1) process, we create a time series of length $N$ by using the relation

$$x(t) = \alpha x(t-1) + \epsilon_t \tag{15}$$

with $\epsilon_t$ a Gaussian random variable with zero mean and constant standard deviation $\sigma_\epsilon = 0.5$. The scaling factor is given as $\alpha = 0.7$ corresponding to the approximate value that we obtained when fitting an AR(1) process to the tree ring width data from eastern Canada (Gennaretti et al., 2014). As initial condition, we use $x(0) = 0.3$. The resulting time series is normalised to zero mean and unit standard deviation. As for GWN, we do not expect to detect significant dynamical anomalies in the corresponding time series.

## 4.3 Non-stationary Rössler system

The Rössler system is defined by the set of ordinary differential equations (ODEs) (Rössler, 1976)

$$\dot{x}(t) = -y(t) - z(t)$$
$$\dot{y}(t) = x(t) + ay(t) \tag{16}$$
$$\dot{z}(t) = b(t) + z(t)(x(t) - c).$$

We here use the two fixed parameters $a = 0.2$ and $c = 5.7$ and a time varying parameter $b(t) = b_0 + \Delta b(t - t_0)$ with $b_0 = 0.02$ and $\Delta b = 0.001$. We numerically solve this system of ODEs with a temporal resolution of $\Delta t = 0.1$ for times in the range $[0, 730]$, discard the first 300 points and then use every seventh point of the remaining time series of the $x$-component to end up with a time series of length $N = 1,000$. As initial conditions we use $x(0) = 0.5$, $y(0) = 0$ and $z(0) = 0$. Again, we normalise the time series to have zero mean and unit standard deviation. The resulting time series and the corresponding Feigenbaum diagram of the stationary system (i. e., $b = $ const.) are shown in the Supplementary Material in Fig. S1.

From the Feigenbaum diagram, it becomes clear that we expect to detect alternating periods of lower and higher dimensional dynamics in the time series. In particular, we stress that we do not expect to detect the bifurcation points but periods of outstandingly high- or low-dimensional dynamics in between them as we use random shuffling surrogates for the pointwise significance test.

## 4.4 Non-stationary Lorenz system

The Lorenz system is given by the following set of ODEs (Lorenz, 1963):

$$\dot{x}(t) = a(y(t) - x(t))$$
$$\dot{y}(t) = x(t)(b(t) - z(t)) - y(t) \tag{17}$$
$$\dot{z}(t) = x(t)y(t) - cz(t).$$





We here use the setting studied in Donges et al. (2011a) and correspondingly fix the parameters $a = 10.0$ and $c = 8/3$, while the parameter $b$ is again varied over time as $b(t) = b_0 + \Delta b(t - t_0)$ with $b_0 = 160.0$ and $\Delta b = 0.02$. We numerically solve

this system of ODEs with a temporal resolution of $\Delta t = 0.05$ for times in the range $[0, 500]$ and use every fifth point of the $x$-component of the system to end up with a univariate time series of length $N = 1,000$. As initial conditions we use $x(0) = 10.0$, $y(0) = 10.0$ and $z(0) = 10.0$. Again, we normalise the time series to have zero mean and unit standard deviation.

The stationary Lorenz system has been found to exhibit a shift from periodic to chaotic dynamics at $b = 166.0$ (Barrio and Serrano, 2007), while for the transient system as described above, transitions could be detected at $b = 161.0$ and $b =$

$166.5$ (Donges et al., 2011a). Thus, we expect to detect regimes of more periodic dynamics for $b < 166.5$ and of more chaotic dynamics for $b \geq 166.5$ in terms of significantly high and low values of the network transitivity, respectively.

### 4.5 Last millennium reanalysis data

Finally, we consider reconstructed temperature and precipitation data of the years $501 - 2000$ AD from the last millennium reanalysis project version 2 (Hakim et al., 2016; Tardif et al., 2019), which combines information from general circulation

models and from proxy measurements using palaeoclimate data assimilation. As no information about the isotopic composition of the precipitation is available, we only consider the models for tree ring width and lake sediments with this input. From the available global gridded data, we use the standardised, thus, unit-less, ensemble average time series of temperature and precipitation from the grid points with the coordinates closest to those for which we have calibrated the tree and the lake model, that is, at $(54° \text{ N}, 70° \text{ W})$.

## 5   Results

### 5.1   Time series properties

As a first step to evaluate the results, we take a look at the properties of the time series generated by the different proxy system models and compare them to the input time series. Supplementary Figures S2 to S4 display the annually sampled input time series for temperature, precipitation and isotopic compositions and the corresponding output time series of the four proxy

system models for the five input scenarios of GWN, the AR(1) process, the non-stationary Rössler system, the non-stationary Lorenz system, and the last millennium reanalysis data. The expected low-pass filter effects of the speleothem, ice and lake models due to the cave residence time, diffusion, and bioturbation, respectively, are directly visible in the time series, while for the tree model, such an effect is neither expected nor visible. Also, it should be noted that the tree ring model with the parameters as specified in Table 1 seems to primarily respond to temperature rather than to precipitation, meaning that the

limiting factor for tree growth in eastern Canada is temperature, which is ecologically reasonable.

For further evaluation, we standardise all time series to zero mean and unit variance and examine some properties of the different input and output series. The left panels of Fig. 1 show the normalised histograms of the input and output variables. To quantify differences in the histograms, we consider the skewness of the distributions of the different time series (see Supple-


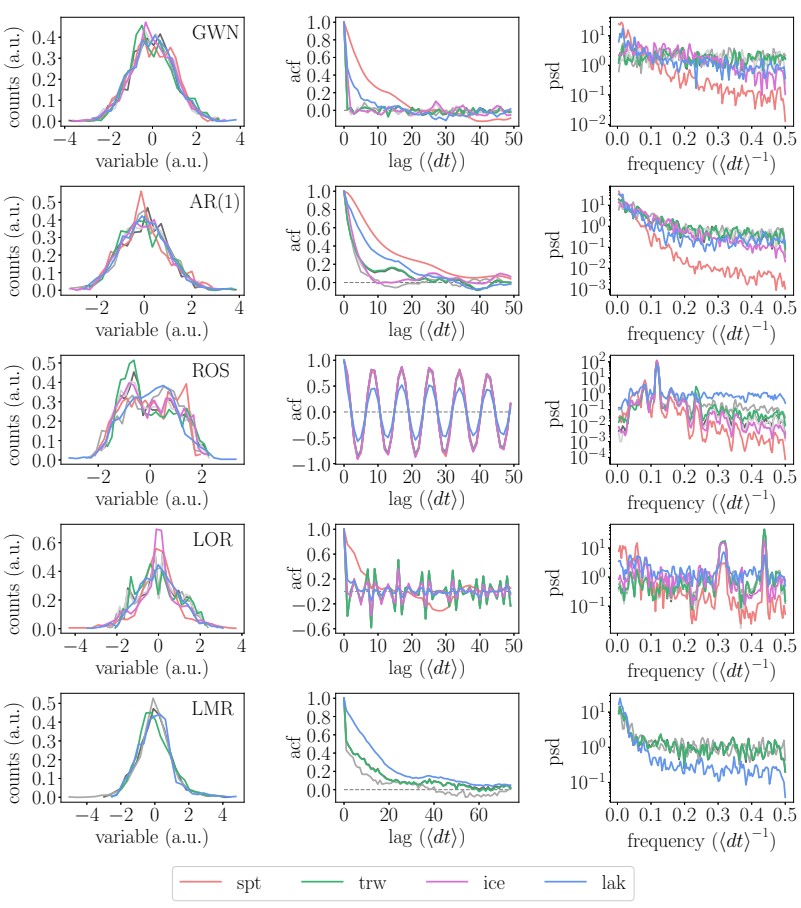

**Figure 1.** Normalised histograms, autocorrelation functions (acf) and estimated power spectral densities (psd) of the different input and proxy system model output time series for GWN, AR(1), ROS, LOR, and LMR (top to bottom). The input variables are denoted in grey, the tree model output in green, the lake model output in blue, the speleothem model output in red, and the ice model output in magenta.

mentary Table S1). We observe that the resulting time series from the tree model all show significant deviations in skewness from the input data, which is probably due to the thresholding of the growth response functions for temperature and soil mois-

ture. For the other proxy models, we observe a significant influence of the model on the skewness of the output for those inputs that are not stochastic. This might either be related to the smaller confidence bounds due to the different surrogate generation, or hint to a different reaction of the models to different distributions of the input data.

Finally, we have a look at the autocorrelation functions and the power spectral densities of the different input and proxy

model output time series. The middle and right panels of Fig. 1 show the resulting autocorrelation functions and estimates of the power spectral density obtained using the Welch method (Welch, 1967). As before, we observe that the autocorrelation function and power spectral density of the tree ring model output closely follow the respective temperature input, while the speleothem model output shows the expected loss of power in the higher frequencies. Similarly, a loss of power in the higher





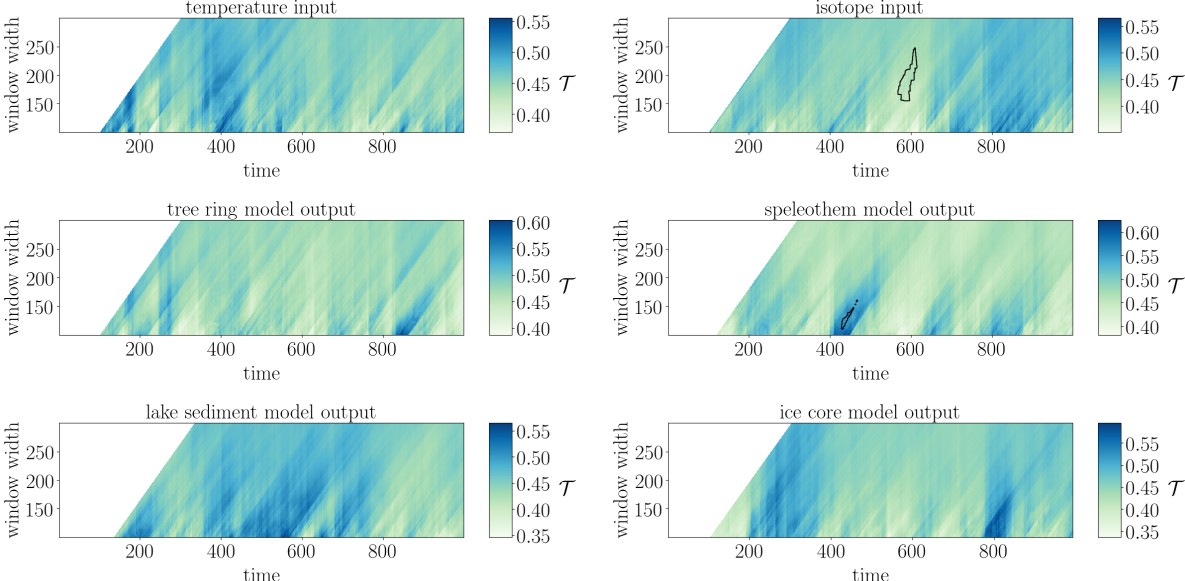

**Figure 2.** Network transitivity (colour-coded) for GWN model in- and output with areawise significance test (contours).

frequencies can be observed in the ice core model output but not as pronounced as for the speleothem model output. The same
holds for the lake sediment model output except for the Rössler and Lorenz scenarios, where the lake model output shows
more power in the higher frequencies than the corresponding input. This is most likely related to the different time-scales of
variability present in those time series in comparison with the others and the particular choice of mixed layer thickness in the
lake sediment model, mxl $= 4$.

## 5.2 Windowed recurrence network analysis

In a second step, we analyse the different proxy model output time series using wRNA as introduced in Sec. 2 and compare the
results to those of the input time series. Figures 2 to 6 show the resulting network transitivity over time for the different input
time series. The analysis is combined with the areawise significance test and areawise significant points indicating anomalously
low or high values of the network transitivity are highlighted by the contours.

For Gaussian white noise (Fig. 2), we do not get any areawise significant points in the input temperature and also the
output of the tree ring width and the lacustrine archive models do not exhibit any areawise significant points. For the isotopic
composition of the precipitation, we find a patch of areawise significant anomalously low values of the network transitivity
which is a random artefact in this particular realisation of GWN. The output of the speleothem and the ice core model do not
show this anomaly, i. e., the processing through the model causes the initially significant (false positive) points to be missed.
Additionally, the speleothem output shows a small patch of falsely identified areawise significant high values of the network
transitivity for small window width after $t = 400$.

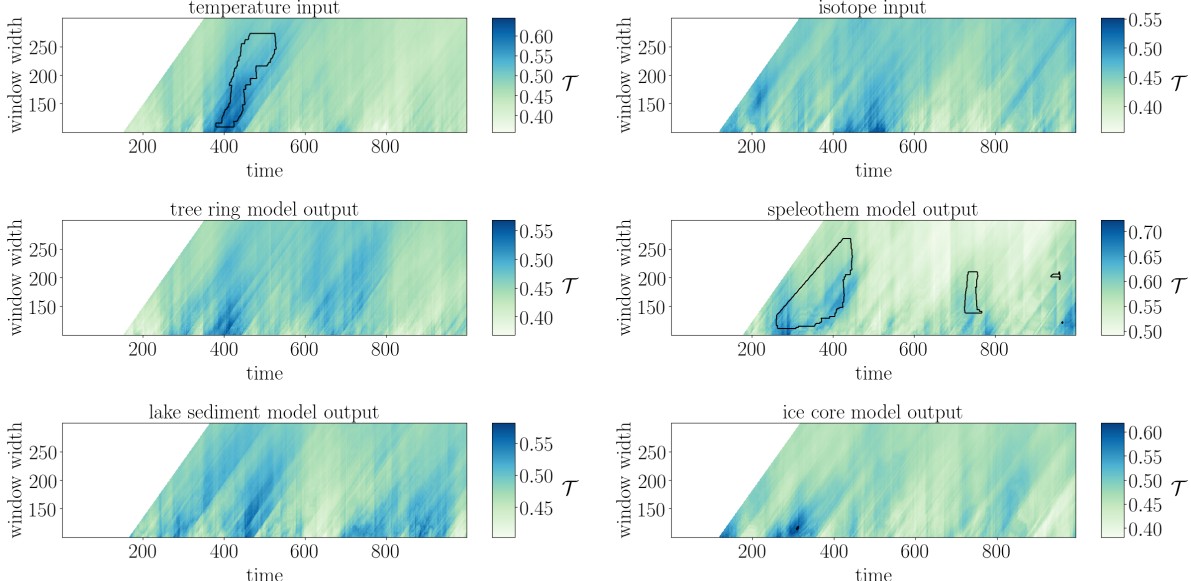

**Figure 3.** Same as in Fig. 2, but for AR(1) model input.

In the case of the AR(1) process (Fig. 3), we observe an areawise significant patch of anomalously high values of network transitivity for the input temperature that is not apparent in the tree ring width and lake sediment model output. For the isotope input, we do not find any areawise significant points, while the speleothem model shows two large and two small falsely identified significant patches of network transitivity.

For the non-stationary Rössler system (Fig. 4), the input time series show areawise significant patches of low network transitivity values in the parameter range $b \in [0.30, 0.35]$ and of high network transitivity for $b \in [0.55, 0.57]$. The isotope input has an additional small areawise significant patch around $b = 0.15$. The tree ring and lake sediment model outputs again show no areawise significant points, i. e., the model processing prevents the significant points to be detected. The speleothem model output only shows the small areawise significant patch at $b = 0.15$ but not the others, while the output from the ice core model

displays the areawise significant patch for $b \in [0.30, 0.35]$.

The input time series of the non-stationary Lorenz system (Fig. 5) exhibits a large patch of areawise significant low values of the network transitivity for high values of $b$ and a small patch of areawise significant high values of the network transitivity around $b = 163.2$. The isotope input shows another small significant patch around $b = 166.6$. The tree ring width output reproduces the large and the small patch of significant values of the network transitivity and additionally exhibits another areawise

significant patch of high transitivity values close to the small patch. The lake sediment output does not show any areawise significant points. The speleothem output shows a large patch of areawise significant high transitivity values for $b \in [162.5, 166.5]$ and, like this, captures the small patch from the isotope input, but also exhibits a large number of falsely identified significant points, while the large patch of low transitivity values from the input is not found to be areawise significant for the model
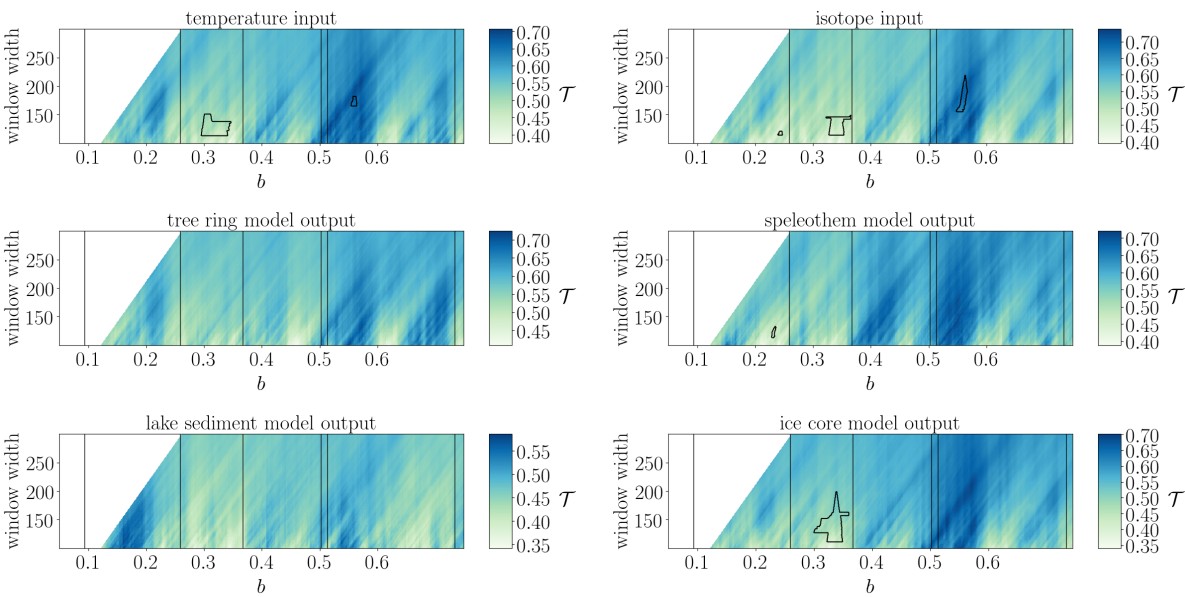

**Figure 4.** Same as in Fig. 2, but for non-stationary Rössler system input. The vertical lines mark the expected bifurcation points identified in the Feigenbaum diagram in Supplementary Fig. S1.

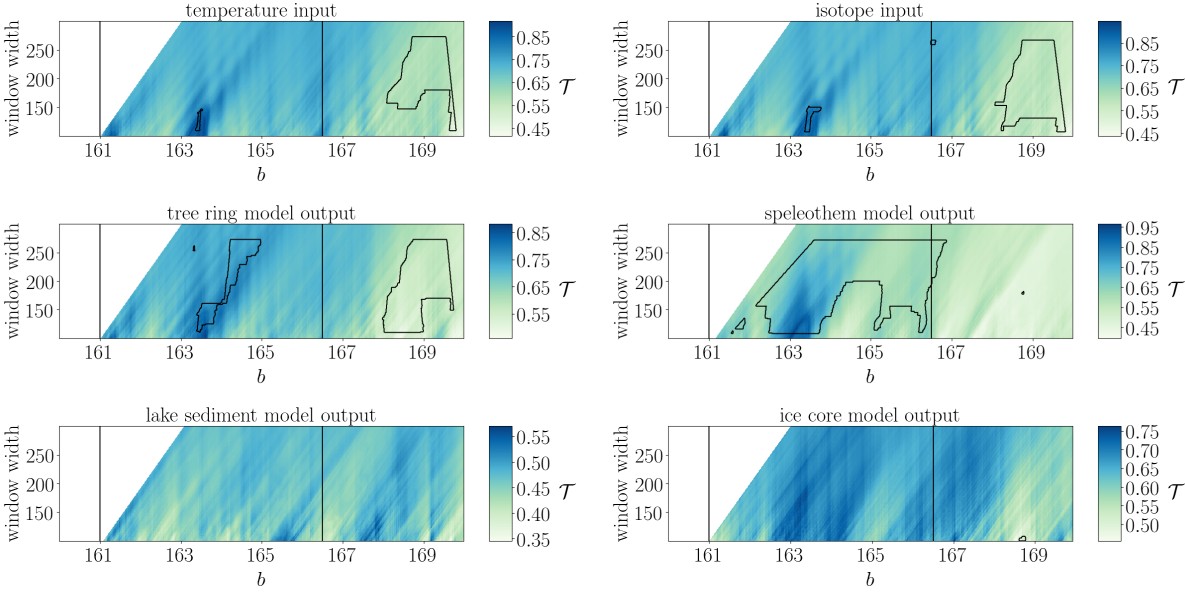

**Figure 5.** Same as in Fig. 2, but for non-stationary Lorenz system input. The vertical lines denote the transitions at $b = 161.0$ and $b = 166.5$ as detected in Donges et al. (2011a).

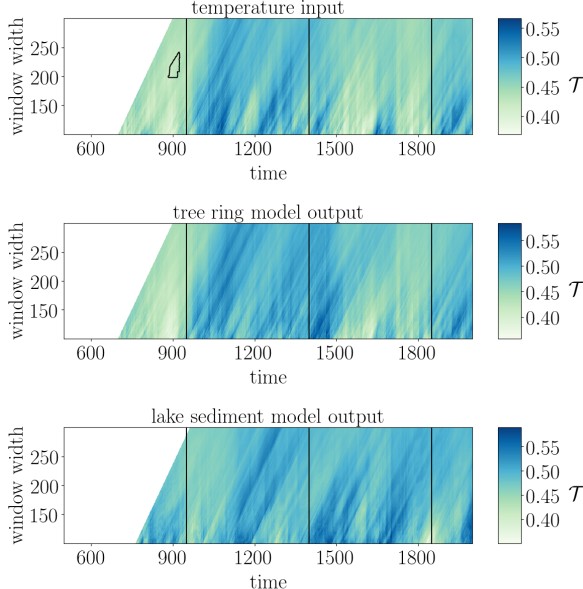

**Figure 6.** Network transitivity (colour-coded) for last millennium reanalysis input at (54° N, 70° W) and corresponding output with areawise significance test (contours). The vertical lines denote the approximate onset dates of the Medieval Climate Anomaly, the Little Ice Age, and the industrial age, respectively.

output except for a very small part of it. The ice core model output only shows a small patch for high values of $b$ and low values

of the window width and misses most of the areawise significant points apparent in the isotope input.

For the last millennium reanalysis data temperature input (Fig. 6), we find a patch of areawise significant values of anomalously low values of the network transitivity around the year 900 AD, roughly coinciding with the onset of the European Medieval Climate Anomaly (denoted by a vertical line at 950 AD). The other vertical lines depict the approximate onset of the European Little Ice Age at 1400 AD, and the onset of the industrial age at 1850 AD. It should be noted that the timings and

imprints of these episodes are known to exhibit substantial regional differences (Franke and Donner, 2017), thus, the reference lines are only for orientation. The model output time series do not show any areawise significant points even though the network transitivity of the tree ring model output varies similarly as the network transitivity of the temperature input with higher values corresponding to lower-dimensional dynamics during the Medieval Climate Anomaly and lower values corresponding to higher-dimensional dynamics during the Little Ice Age.

In order to quantify the effect of the proxy system models as nonlinear filters of the input signal on the detection of areawise significant points, Table 5 displays the fraction of falsely identified and missed significant points in the different proxy models for the different input time series. We observe that for the tree ring width model, the lacustrine sediment model, and the ice core model missed significant points are more common than falsely identified significant points, with the exception of the Lorenz input for the tree ring width model. For the speleothem model, both falsely identified and missed significant points occur. In





**Table 5.** Fraction of missed and falsely identified significant points in the different proxy models with respect to the corresponding reference input variables.

| archive | reference | | GWN | AR(1) | ROS | LOR | LMR |
|---------|-----------|---|-----|-------|-----|-----|-----|
| tree rings | temperature | missed | 0.000 | 0.056 | 0.012 | 0.003 | 0.004 |
| | | falsely identified | 0.000 | 0.000 | 0.000 | 0.064 | 0.000 |
| lake sediments | temperature | missed | 0.000 | 0.056 | 0.011 | 0.083 | 0.004 |
| | | falsely identified | 0.000 | 0.000 | 0.000 | 0.000 | 0.000 |
| speleothems | isotopes | missed | 0.010 | 0.000 | 0.011 | 0.106 | – |
| | | falsely identified | 0.001 | 0.105 | 0.001 | 0.289 | – |
| ice cores | isotopes | missed | 0.010 | 0.000 | 0.006 | 0.109 | – |
| | | falsely identified | 0.000 | 0.000 | 0.011 | 0.001 | – |

fact, this model shows particularly high fractions of falsely identified and missed significant points, while overall, the ice core model seems to best reproduce the results of the input.

Taken together, the previous findings should raise awareness in the context of future applications of wRNA to palaeoclimate proxy time series, suggesting that interpretations of results obtained for individual records only may not be sufficiently robust for drawing substantiated conclusions. From a practical perspective, this calls for combining different time series from different proxies and/or archives from the same region to obtain further climatological knowledge from such kind of analysis (Donges et al., 2015a; Franke and Donner, 2017).

## 6   Discussion and conclusions

In this paper, we have studied the suitability of windowed recurrence network analysis (wRNA) for detecting dynamical anomalies in time series from different proxy archives. For this, we used proxy system models that simulate the formation of proxy archives, such as for example tree rings, lake sediments, speleothems, and ice cores, given some climatic input variables like temperature and precipitation. We created artificial input time series with different properties and additionally used temperature and precipitation data from the last millennium reanalysis project (Hakim et al., 2016; Tardif et al., 2019). We then processed the input time series through the different proxy models and compared time series properties such as the autocorrelation function for the input and model output time series. Finally, we contrasted the results of wRNA for the different input and model output time series.

We observed the expected loss of power in the higher frequencies for the lake sediment, speleothem, and ice core model output time series (due to intrinsic smoothing/integration processes inherent to the physical system) and noted changes in the skewness of the model output time series for the deterministic and real-world input scenarios. The tree ring width model showed changes in skewness also for the stochastic input. For the wRNA, we found that time series of tree ring width and brGDGTs in lake sediments have problems with missing areawise significant points, while the isotopic composition of speleothems also





exhibits falsely identified significant points. Time series of the isotopic composition of ice yield comparable results to the corresponding input but also sometimes miss significant points.

For the stochastic input time series as for example the isotope input of GWN, we found some areawise significant artefacts in single realisations. To improve the reliability of the results for the these processes, more realisations should be considered

to confirm the results and to exclude the influence of random artefacts. As we applied an areawise significance test to identify dynamical anomalies which reduces the number of false positives in the analysis results, this can also reduce the number of true positives and increase the number of false negatives independent of whether considering model input or output time series. In this regard, we also observed that time series with stronger autocorrelations in most cases show higher correlations for the wRNA results in the different domains and, thus, have more restrictive bounds of the areawise significance test (not shown).

Falsely identified significant points in the model output, particularly present in the results for the speleothem model, might for the stochastic processes be related to the strong filtering within the model. For the Lorenz input, the processing through the speleothem model leads to large fractions of both, falsely identified and missed significant points as the high values of the network transitivity before the transition at $b = 166.5$ are classified as significant, while in the input, the low values of the network transitivity after $b = 166.5$ are classified as significant. Missed significant points in the model output may also be

related to the processing through the model destroying some structure within the data such as the thresholding process in the tree ring width model.

Future work should include the study of alternative proxy system models within this framework. Results of proxy system models for both, the same proxies (but with more detailed systemic understanding of the formation of the proxies) and different proxy variables (in particular, for other lacustrine proxy variables), will complement the improved understanding of the suit-

ability of wRNA for these types of time series and will advance the interpretation of the corresponding results. Also, sensitivity studies for the different model parameters are of interest to better interpret results obtained with wRNA for a given real-world data set. This concerns particularly the mean aquifer transit time of the speleothem model.

Additionally, the study of properties of the analysed time series can serve as starting point to judge the suitability of wRNA for other data to be analysed. In particular, the effect of filtering the time series with different non-linear filters prior to the

analysis as done within the different proxy system models can and should be studied more systematically. In particular, the theory of non-linear observability might give an interesting new perspective on this as the filtering can be seen as creating a new observable, and the choice of observable has already been shown to influence results of recurrence quantification analysis and recurrence network analysis (Portes et al., 2014, in press). Moreover, further systematically studying the relation between the autocorrelation of a time series and the resulting network properties might yield additional information on the role of the

different archives for wRNA.

*Code availability.* Exemplary Python code for the windowed recurrence network analysis including the areawise significance test is available at https://gitlab.pik-potsdam.de/lekscha/awsig. A comprehensive implementation of recurrence network analysis can be found in the open-source Python package `pyunicorn` (Donges et al., 2015b), which can be found at https://github.com/pik-copan/pyunicorn.



*Author contributions.* JL and RVD designed the research. JL performed the numerical experiments and data analyses. JL and RVD discussed the results and wrote the manuscript.

*Competing interests.* The authors declare no competing interests.

*Acknowledgements.* This work has been financially supported by the German Federal Ministry for Education and Research (BMBF) via the BMBF Young Investigators Group *CoSy-CC$^2$ - Complex Systems Approaches to Understanding Causes and Consequences of Past, Present and Future Climate Change* (grant no. 01LN1306A) and the Studienstiftung des deutschen Volkes. Calculations have been performed with the help of the Python package `pyunicorn` (Donges et al., 2015b).





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
