# Peer review of "Detecting dynamical anomalies in time series from different palaeoclimate proxy archives using windowed recurrence network analysis"

_Nonlinear Processes in Geophysics, 2019_

## Referee Comment (RC1) · Dmitry Divine (Referee) · 16 Oct 2019

**Review of a manuscript for *NPG***

**Detecting dynamical anomalies in time series from different palaeoclimate proxy archives using windowed recurrence network analysis** by J. Lekscha and R. Donner

**Overall:**

In this manuscript the authors test the suitability of the earlier developed windowed recurrence network analysis (wRNA) for detecting dynamical anomalies in paleoclimate proxy times series. The method skills are tested on the suite of stationary and nonstationary timeseries of forward modelled pseudoproxies with known dynamical properties. This work is a natural continuation/extension of the earlier studies of the group on the application of networks to the analysis of (paleo)climatic data

The paper is clearly written and results are well presented. I therefore consider the manuscript deserves to be published after some very minor modifications /additions to the content if the authors/editor finds them relevant.

**Minor comments**

Page 3 Line 59: "for estimating embedding delay…autocorrelation function" is it a global or a windowed estimate? Please be specific

Page 3 Line 66: why namely the maximum norm is used? Is it possible to justify the choice? Did the authors check the sensitivity of the results to the use of other norms?

Page 3 Line 71: "…such that a fraction \rho of all possible links in the network is realized": is the threshold global or window-based?

Page 3 Eq. 4 please indicate that |i-j|=|v-i|=1

Page 5-6: forward proxy model for tree rings. One should not that the juvenile growth is not modelled/accounted for in the model used. Hence an effect of its subtraction, which can be substantial, depending on the species used, technique applied and the entire age structure of the tree-ring network (archive) is also discarded. It is worth mentioning in a context of results demonstrated for tree rings.

Page 8 table 2: Please check if the amount of measured foraminifera is correct (number of species? Sample weight?) please indicate units

Page 11: Use of nonstationary Røssler system: How realistic this model actually is for climate applications? Are there any larger-scale climatic processes that can potentially be associated with this model?

Page 12 Line 309: "…respond to temperature rather than to precipitation…" mind that compared with a temperature, precipitation is not reproduced in the models that well, though for this particular case (boreal forest), temperature indeed will be a stronger limiting factor.

Page 13 Line 322: "…closely follow the respective temperature input", note my comment on the used forward proxy model for tree rings. Such a good concistency can partly be attributed to a lack of juvenile growth effect in the model.

Page 17 Line 368: "….lower-dimensional dynamics during the MCA…..higher-dimensional… during the LIA" Can the authors elaborate a bit more on this result? What are the actual features in the analyzed timeseries manifested in wRNA as lower and higher network \Tau?

Page 19 Lines 403-404: Did the authors consider block shuffling of surrogates (same as in block bootstrapping) as a possible method to tackle this problem?

---

## Referee Comment (RC2) · Michel Crucifix (Referee) · 20 Nov 2019

First of all, I would like to apologise for this delayed review.

The purpose of the manuscript is to apply a method developed and documented by the authors elsewhere (the windowed recurrence network analysis (wRNA)) on artificial time series simulating the output of speleothem, lake, tree archives and isotopic water concentration in ice cores. The purpose is to test to what extent the 'proxy process' (transformation of the climate signal by the natural archive) could mask anomalies that

would have been otherwise detected in the original time series. Different time series have been tested, including Gaussian white noise, an autoregressive process of order 1, output non-linear dynamical system model, and data from the reanalysis project of Hakim et al.

The main diagnostic is 'network transitivity', and the application of the method to the different input datasets generates Figures 2 to 6. The reader is invited to concentrate on the 'area-wise' significance test, which is supposed to indicate a signal of significant change in network transitivity, to be interpreted as a change in the effective dimensionality of the system.

Some results appear a bit disconcerting, especially the test on the AR(1) signal, because it shows, on the one hand, a large patch of area-wise significant anomalies in network transitivity which was — if I understood correctly — a priori not expected in this signal. In addition, this large patch disappears in the natural archives simulated with this model, while another patch emerges in the speleothem simulation. Simulations with other inputs tend to confirm that the speleothem model is prone to create or destroy areas of significant network transitivity anomalies seen in the original time series.

The study is quite systematic in its approach, to the point of being slightly repetitive, and yet, one might argue that it falls short convincing the reader about the robustness of the conclusions. Basically, up to p. 12 the manuscript consists in an exposition of the methods, which for their greatest part have been described elsewhere (the significant area test is in press, and the proxy models have been published elsewhere). The output of the wRNA analysis follows a show-and-tell description running until p. 18, and even though some main conclusions are correctly outlined, the discussion does not really help identify mechanisms or key conclusions that would actually help to 'improve the interpretation of windowed recurrence network analysis' as announced in the abstract. For example, the authors have observed that the speleothem model modifies the significant patches, but we do not really which process, in the speleothem model,

is responsible for this behaviour. Do we expect this to be an idiosyncrasy of the particular speleothem model used here, or do we expect it to be a general result? Which aspects of the 'nonlinear filtering' should be incriminated? The presence of a large significant area patch on the AR(1) time series, along with its the quasi-absence of significant changes in network transitivity in the last-millennium reanalysis data is also disconcerting, because we no longer know how to reasonably interpret the output of the wRNA for understanding climate dynamics. Is the last millennium actually the right test bed for this study?

To sound hopefully a bit more constructive, I would suggest the authors to seek for more general aspects of the filtering process which may destroy or generate spurious changes in network transitivity. Is this caused by non-linearity in the instantaneous response (what would an 'exp(x)' filtering generate?) Is this the temporal smoothing process? Is it the amount of noise? What would this analysis tell us about how to find proxies that would preserve the wRNA signal, beyond the particular example chosen here? Which are the desirable characteristics for such proxies? Answering these questions would provide some more general and perhaps valuable hints for the interpretation of the wRNA, which could then be summarised in the abstract.

Perhaps the reader will also better appreciate the interest of the wRNA if more clues are given about how to interpret it: can one get a more or less adequate intuition of what a change in wRNA implies about the dynamics of climate. What, physically, does an increase or a decrease in network transitivity mean? Would this be associated with a form of 'global synchronisation' ? Are we expecting it when we approach a form of bifurcation (a "tipping point")? What is the wRNA telling us that is not obvious from visual inspection of the time series?

Finally, the choice of an embedding dimension $m = 3$ was, to this reader, difficult to reconcile the quote that "The embedding theorem of Takens guarantees that, when choosing the embedding dimension larger than twice the box-counting dimension of the original attractor, the reconstructed and the original system's attractor are related by a

smooth one-to-one coordinate transformation with smooth inverse, independent of the choice of the delay". Wouldn't we have expected, on this basis, a much larger embedding dimension? This may invite some discussion, perhaps available in Lekscha and Donner, (in press). In 1984 (Nature, vol. 311, p. 311), Nicolis and Nicolis published an estimate of the 'climate attractor dimension' but subsequent authors (including Grassberger, 1986, Nature, 1996, vol. 323, p. 609, and Vautard and Ghil, 1989, Physica D, vol. 35, p.395) pointed the difficulty of actually getting a meaningful estimate of "a climate dimension" from a 1-dimensional, finite record. Could the authors clarify their position in this respect?

---

## Author Response (AR1)

Dear Editor,

With this letter we would like to resubmit our manuscript entitled "Detecting dynamical anomalies in time series from different palaeoclimate proxy archives using windowed recurrence network analysis" for publication in *Nonlinear Processes in Geophysics.*

In the revision, we implemented all changes that we mentioned in our comments during the discussion phase. In the following, we reproduce our answers to the Referee's comments to our manuscript (corresponding to the answers provided during the discussion phase) and highlight the associated changes in the revised manuscript. A difference of the originally submitted and resubmitted manuscript can be found at the end of this document.

Yours sincerely,
Jaqueline Lekscha and Reik Donner

**Answers to Referee 1**

**Overall:**
**In this manuscript the authors test the suitability of the earlier developed windowed recurrence network analysis (wRNA) for detecting dynamical anomalies in paleoclimate proxy times series. The method skills are tested on the suite of stationary and nonstationary timeseries of forward modelled pseudoproxies with known dynamical properties. This work is a natural continuation/extension of the earlier studies of the group on the application of networks to the analysis of (paleo)climatic data**

**The paper is clearly written and results are well presented. I therefore consider the manuscript deserves to be published after some very minor modifications /additions to the content if the authors/editor finds them relevant.**

We thank the Referee for this positive assessment of our work. Our replies to the suggested modifications and additions are given below.

**Minor comments**
**Page 3 Line 59: "for estimating embedding delay...autocorrelation function" is it a global or a windowed estimate? Please be specific**

It is a global estimate. We have specified this in the revised version of the manuscript (line 64).

**Page 3 Line 66: why namely the maximum norm is used? Is it possible to justify the choice? Did the authors check the sensitivity of the results to the use of other norms?**

We use the maximum norm because there is a direct analytical relationship between the network transitivity and the dimensionality of the dynamics of the system for it, as shown in Donner et al., EPJB, 2011. Thus, using the maximum norm is particularly useful when assessing the network transitivity to study the system's dynamics. Moreover, due to its specific form, the calculation of the maximum norm of a given vector in Euclidean space is particularly simple. This is also why this specific norm has been particularly widely used in previous works on recurrence network analysis and other recurrence plot based techniques. We have clarified this in the revised manuscript (lines 72/73 and 83/84).

In our present work, we did not explicitly check the sensitivity of the results with respect to the use of other norms for the sake of conciseness. However, we performed similar sensitivity tests for other (similar) time series and typically found the results to be qualitatively robust when, for example, using the Euclidean norm instead of the maximum norm. We expect this to also hold for the present work.

**Page 3 Line 71: "...such that a fraction \rho of all possible links in the network is realized": is the threshold global or window-based?**

For each window, we construct a recurrence network and thus, set the threshold \epsilon for every window separately. That is, in each window, a fraction \rho of all possible links are realized. We have clarified this in the revised manuscript (lines 76/77).

**Page 3 Eq. 4 please indicate that |i-j|=|v-i|=1**

In this equation, the summation is performed over all i, j and v and not only over neighboring indices. In fact, the network transitivity as given by Eq. 4 can be interpreted as the probability that two random neighbors of a randomly chosen node are mutually connected. Assuming that the randomly chosen node has index v, the denominator of Eq. 4 counts how many combinations of all i and j are neighbors of v. By summing over all possible v, the denominator equals the number of all possible triangles in the network. The nominator then counts how many of the possible triangles are actually realised.

**Page 5-6: forward proxy model for tree rings. One should not that the juvenile growth is not modelled/accounted for in the model used. Hence an effect of its subtraction, which can be substantial, depending on the species used, technique applied and the entire age structure of the tree-ring network (archive) is also discarded. It is worth mentioning in a context of results demonstrated for tree rings.**

We thank the Referee for drawing our attention to this important point. Indeed, the model does not take into account juvenile growth and it would be very interesting to compare the results to those of a model that does account for this effect in future work. We have added a corresponding comment in the revised manuscript (lines 149-151 and 412/413).

**Page 8 table 2: Please check if the amount of measured foraminifera is correct (number of species? Sample weight?) please indicate units**

For this model, we used the default values for the amount of measured foraminifera, mixed layer thickness, and abundance of species of the TURBO2 model which are unitless (compare Trauth, Comp. Geosci., 2013). Of course, it would be very interesting to compare the results for varying values of these parameters. Unfortunately, such a systematic study on the effects of different model parameter settings was beyond the scope of our present manuscript and is thus left as a topic for future work. Generally, we expect here more quantitative rather than qualitative changes if the mentioned parameters are varied within some reasonable ranges.

**Page 11: Use of nonstationary Røssler system: How realistic this model actually is for climate applications? Are there any larger-scale climatic processes that can potentially be associated with this model?**

Unlike the Lorenz system, which has been originally introduced as a simplified toy model for atmospheric convection processes, the Rössler system has no such close climatological interpretation to the best of our knowledge. One of the main motivations for introducing this model

(inspired by chemical reaction kinetics in Rössler's (Phys. Lett. A, 1976) original work) was to provide a chaotic dynamical system model with somewhat simpler behavior than the Lorenz system (i.e., without a double-scroll structure). This simpler type of attractor topology, along with the still non-trivial and rich cascade of bifurcations, was the main motivation for us to use the Rössler system as a generator for complex input dynamics to our proxy system models. Notably, we expect that this well studied model is known to a vast majority of the readership of Nonlinear Processes in Geophysics. By contrast, we did not attempt here to reach any dedicated level of realism in making the input variable particularly similar to real world climate dynamics. We have clarified this point in our revised manuscript (lines 262/263 and 268).

**Page 12 Line 309: "...respond to temperature rather than to precipitation..." mind that compared with a temperature, precipitation is not reproduced in the models that well, though for this particular case (boreal forest), temperature indeed will be a stronger limiting factor.**

As we chose the model parameters for the tree ring width model to simulate tree growth in Eastern Canada, we were indeed expecting that the temperature and not the precipitation will be the limiting factor for the model in this case. However, we fully agree with this comment, and added a brief explicit statement on this fact in the revised supplement (lines 9-13).

**Page 13 Line 322: "...closely follow the respective temperature input", note my comment on the used forward proxy model for tree rings. Such a good concistency can partly be attributed to a lack of juvenile growth effect in the model.**

We agree with the Referee that the good correspondence between the temperature input and the tree ring width model output can partly be attributed to the lack of correction for juvenile growth in the tree ring width model. We have commented on this in the revised manuscript (lines 149-151 and 412/413).

**Page 17 Line 368: "....lower-dimensional dynamics during the MCA.....higher-dimensional... during the LIA" Can the authors elaborate a bit more on this result? What are the actual features in the analyzed timeseries manifested in wRNA as lower and higher network \Tau?**

The MCA has often be attributed to more stable climate conditions while the LIA has often been attributed to more variable climate conditions even though this imprint has varied locally and has been mainly discussed for Europe. As described, for the LMR data from Eastern Canada, we do not find any significant periods in the model output data meaning that we cannot detect any dynamical anomalies in the model output. Still, for the tree ring width model output, we observe higher values of the network transitivity roughly coinciding with the MCA and lower values of the network transitivity during the LIA. As the network transitivity ($\mathcal{T}$) has been shown to be related to some proxy for the dimensionality of the system's dynamics (m) via the relation $m=\log(\mathcal{T}) / \log(3/4)$ (cf. Donner et al., EPJB, 2011), we tentatively conclude that the higher values of the network transitivity during the MCA and the lower values of the network transitivity during the LIA reflect a lower/higher dimensional dynamics of the system at this particular location (in terms of complexity of temporal variations rather than just a change in variance). In terms of the time series properties for the different periods, we indeed additionally observe an increase in variance of the time series from the MCA to the LIA, which is very likely also reflected in the different recurrence networks and, thus, the resulting network transitivity. We note that this non-stationarity in variance along with the MCA-LIA transition in the European/North Atlantic sector has also been reported as being reflected in other nonlinear characteristics, which have been previously interpreted as a hallmark of some dynamical anomaly (Schleussner et al., Clim. Dyn., 2015; Franke & Donner, Clim. Change, 2017).

We have added a corresponding more detailed discussion of the results to the revised manuscript (lines 355-366).

**Page 19 Lines 403-404: Did the authors consider block shuffling of surrogates (same as in block bootstrapping) as a possible method to tackle this problem?**

We did not consider block shuffling of surrogates in this work, but rather addressed this issue by using iterative amplitude adjusted Fourier transform surrogates (that is, surrogates that exactly preserve the probability distribution and linear correlation structure of the data) for the areawise significance test.

We thank the Referee for suggesting using block shuffling as a possibly less computationally demanding alternative and will keep this option in mind for further work on significance tests. For the present work, we think that adding an additional type of surrogate data would primarily increase the already large amount of material presented without providing topically relevant results markedly deviating from those reported. Still, we agree that this is a very important topic, and suggest that apart from block shuffling surrogates, also other surrogate routines should be studied more systematically for time series with different autocorrelation properties.

**References**

Donner, R. V., Heitzig, J., Donges, J. F., Zou, Y., Marwan, N., and Kurths, J.: The geometry of chaotic dynamics — a complex network perspective, The European Physical Journal B, 84, 653–672, 2011.

Franke, J. G., Donner, R. V.: Dynamical anomalies in terrestrial proxies of North Atlantic climate variability during the last 2ka, Climatic Change, 143(1-2), 87-100, 2017.

Rössler, O.: An equation for continuous chaos, Physics Letters A, 57, 397 – 398, 1976.

Schleussner, C.-F., Divine, D., Donges, J. F., Miettinen, A., Donner, R. V.: Indications for a North Atlantic ocean circulation regime shift at the onset of the Little Ice Age, Climate Dynamics, 45, 3623-3633, 2015.

Trauth, M. H.: TURBO2: A MATLAB simulation to study the effects of bioturbation on paleoceanographic time series, Computers & Geosciences, 61, 1 – 10, 2013.

**Answers to Referee 2**

**The purpose of the manuscript is to apply a method developed and documented by the authors elsewhere (the windowed recurrence network analysis (wRNA)) on artificial time series simulating the output of speleothem, lake, tree archives and isotopic water concentration in ice cores. The purpose is to test to what extent the 'proxy process' (transformation of the climate signal by the natural archive) could mask anomalies that would have been otherwise detected in the original time series. Different time series have been tested, including Gaussian white noise, an autoregressive process of order 1, output non-linear dynamical system model, and data from the reanalysis project of Hakim et al.**

**The main diagnostic is 'network transitivity', and the application of the method to the**

**different input datasets generates Figures 2 to 6. The reader is invited to concentrate on the 'area-wise' significance test, which is supposed to indicate a signal of significant change in network transitivity, to be interpreted as a change in the effective dimensionality of the system.**

**Some results appear a bit disconcerting, especially the test on the AR(1) signal, because it shows, on the one hand, a large patch of area-wise significant anomalies in network transitivity which was — if I understood correctly — a priori not expected in this signal. In addition, this large patch disappears in the natural archives simulated with this model, while another patch emerges in the speleothem simulation. Simulations with other inputs tend to confirm that the speleothem model is prone to create or destroy areas of significant network transitivity anomalies seen in the original time series.**

Indeed, the AR(1) signal shows a patch of areawise significant anomalies that is a priori not expected. As argued in the manuscript (original manuscript, lines 398-400), such artefacts appearing in single realisations of the stochastic input time series should be excluded in future work by considering ensembles of realisations of the processes. Unfortunately, this was not possible within the scope of the original manuscript.

With respect to the results of the different proxy model output time series, we generally found that the tree ring and the lake sediment model tend to miss, while the speleothem model both misses and creates additional areawise significant patches of the network transitivity, which is in agreement with the above observations. We have further disentangled the description of these results in the revised manuscript in order to make these points clearer (lines 313-366).

**The study is quite systematic in its approach, to the point of being slightly repetitive, and yet, one might argue that it falls short convincing the reader about the robustness of the conclusions. Basically, up to p. 12 the manuscript consists in an exposition of the methods, which for their greatest part have been described elsewhere (the significant area test is in press, and the proxy models have been published elsewhere). The output of the wRNA analysis follows a show-and-tell description running until p. 18, and even though some main conclusions are correctly outlined, the discussion does not really help identify mechanisms or key conclusions that would actually help to 'improve the interpretation of windowed recurrence network analysis' as announced in the abstract. For example, the authors have observed that the speleothem model modifies the significant patches, but we do not really which process, in the speleothem model, is responsible for this behaviour. Do we expect this to be an idiosyncrasy of the particular speleothem model used here, or do we expect it to be a general result? Which aspects of the 'nonlinear filtering' should be incriminated? The presence of a large significant area patch on the AR(1) time series, along with its the quasi-absence of significant changes in network transitivity in the last-millennium reanalysis data is also disconcerting, because we no longer know how to reasonably interpret the output of the wRNA for understanding climate dynamics. Is the last millennium actually the right test bed for this study?**

We thank the Referee for pointing out the weaknesses of the presentation of the material and have worked on the mentioned points in the revised manuscript. This particularly concerns the presentation of the results and the corresponding discussion and conclusions (lines 313-366 and 387-424). We also decided to move the description of the time series properties of the model output time series to the supplementary information document. For the (admittedly long) description of the methods and proxy system models, we consider it useful to be included in the manuscript to provide

a complete picture and make the contribution self-consistent. Still, we shortened the detailed description of the areawise significance test (lines 109-112).

Also, we want to stress that this study has been meant to improve the interpretation of wRNA results in terms of highlighting not only the potentials, but also the limitations of the method when applied to palaeoclimate data. That is, in particular for the speleothem model, our results show the need of further studying the effects of different filtering mechanisms on the results of the wRNA in order to draw reliable conclusions when analysing real-world data. (Note that there have been quite a few papers reporting the results of windowed recurrence analyses of real-world speleothem data, e.g., Donges et al., Clim. Past, 2015; Eroglu et al., Nat. Comm., 2016.) In this regard, we argue that the last millennium is a good test bed because for this period, highly resolved proxy data are available from various different archives. We have clarified the direct conclusions from the obtained results with respect to the interpretation of the wRNA and outline further work that will help to increase the robustness of the conclusions (lines 397-403). For the speleothem model results, for example, we expect the conclusions to hold, independently of the actual model used, but potentially depend on the choice of the model parameters.

**To sound hopefully a bit more constructive, I would suggest the authors to seek for more general aspects of the filtering process which may destroy or generate spurious changes in network transitivity. Is this caused by non-linearity in the instantaneous response (what would an 'exp(x)' filtering generate?) Is this the temporal smoothing process? Is it the amount of noise? What would this analysis tell us about how to find proxies that would preserve the wRNA signal, beyond the particular example chosen here? Which are the desirable characteristics for such proxies? Answering these questions would provide some more general and perhaps valuable hints for the interpretation of the wRNA, which could then be summarised in the abstract.**

We thank the Referee for these suggestions and fully agree that including the results for a set of general filtering functions and noises on the wRNA results will help interpreting the obtained results by providing insights into the mechanisms that are responsible for them in the different proxy system models. We have performed some further analyses and briefly elaborated on these more general aspects in the revised version of the manuscript (lines 304-312).

**Perhaps the reader will also better appreciate the interest of the wRNA if more clues are given about how to interpret it: can one get a more or less adequate intuition of what a change in wRNA implies about the dynamics of climate. What, physically, does an increase or a decrease in network transitivity mean? Would this be associated with a form of 'global synchronisation' ? Are we expecting it when we approach a form of bifurcation (a "tipping point")? What is the wRNA telling us that is not obvious from visual inspection of the time series?**

We agree with the Referee that including a paragraph on the interpretation of wRNA with respect to the climate system is a very good idea. Still, general statements will hardly be possible as climate-related interpretations vary depending on the location and thus, local boundary conditions have to be taken into account. Also, the network transitivity calculated from a single time series cannot be associated with a 'global synchronisation' as only information from a single location is taken into account. Instead, the network transitivity has rather been related to the dynamical regularity of the variations in the analysed time series (e.g., Donges et al., PNAS, 2011) with higher values of the network transitivity corresponding to less irregular variability and vice versa. This is in accordance with the interpretation of the network transitivity as an indicator of the dimensionality of the system's dynamics. In this regard, detected anomalies in the network transitivity could be related to some tipping point, but do not have to be. We have attempted to provide a more concise description

of possible interpretations of the wRNA results in the (palaeo)climate context in the revised version of the manuscript (lines 404-410).

**Finally, the choice of an embedding dimension m = 3 was, to this reader, difficult to reconcile the quote that "The embedding theorem of Takens guarantees that, when choosing the embedding dimension larger than twice the box-counting dimension of the original attractor, the reconstructed and the original system's attractor are related by a smooth one-to-one coordinate transformation with smooth inverse, independent of the choice of the delay". Wouldn't we have expected, on this basis, a much larger embedding dimension? This may invite some discussion, perhaps available in Lekscha and Donner, (in press). In 1984 (Nature, vol. 311, p. 311), Nicolis and Nicolis published an estimate of the 'climate attractor dimension' but subsequent authors (including Grassberger, 1986, Nature, 1996, vol. 323, p. 609, and Vautard and Ghil, 1989, Physica D, vol. 35, p.395) pointed the difficulty of actually getting a meaningful estimate of "a climate dimension" from a 1-dimensional, finite record. Could the authors clarify their position in this respect?**

We are well aware of the problem of choosing an appropriate embedding dimension when the available data are univariate, finite, and subject to noise. The embedding theorem of Takens is a sufficient condition and not a necessary one; thus, embedding dimensions smaller than twice the box-counting dimension of the original system's attractor may lead (at least approximately) to an appropriate embedding, which might possibly, at least partly, reconcile the quote and the choice of the embedding dimension m = 3. Also, it should be noted that the theorem is strictly valid only for perfect data. Thus, for finite and noisy data, the estimation of the embedding dimension has to rely on some more or less heuristic approaches. Many of these approaches however either have problems of distinguishing deterministic chaos and noise or systematically underestimate the embedding dimension.

In this spirit, we fully agree with the critiques of the cited papers that it is difficult to get a meaningful estimate of an embedding dimension from univariate, finite and noisy data. For the particular case of a 'climate attractor dimension', we agree that the climate system is not a low-dimensional dynamical system. Still, we think that lower dimensional embeddings can be used to obtain meaningful information about a system. Furthermore, in the palaeoclimate context where available time series are often rather short, we do not think that high-dimensional embeddings are useful as the limited amount of data points will in most cases not be sufficient to sample the attractor in a high-dimensional embedding space.

In the majority of the paper, we do not directly analyse climate data but rather synthetic data representing different kinds of underlying processes. For convenience, comparability, and with respect to the time series length and the increasing computational effort for larger embedding dimensions, we chose m = 3 for all analysed time series, thus, also for the last millennium reanalysis data which indeed represents more closely the actual dynamics of the climate system. In general, when analysing real-world data, we think that the problem of choosing an appropriate embedding dimension is best tackled by employing one of the estimation methods that can distinguish deterministic and stochastic signals and do not require too many subjective parameter choices (such as the one presented in Cao (1997), Physica D, 110, 43-50), and additionally slightly varying the obtained embedding dimension when analysing the data to check the robustness of the results. In the revised manuscript, we added a more detailed comment on our choice of m = 3 (lines 55-64).

[revised manuscript text omitted]

**Time series properties**

In order to complement the results of recurrence network analysis and better understand their possible dynamical meaning, we take a look at the properties of the time series generated by the different proxy system models and compare them to those of the input time series. Figures S2 to S4 display the annually sampled input time series for temperature, precipitation and isotopic compositions and the corresponding output time series of the four proxy system models for the five input scenarios of GWN, the AR(1) process, the non-stationary Rössler system, the non-stationary Lorenz system, and the last millennium reanalysis data. The expected low-pass filter effects of the speleothem, ice and lake models due to the cave residence time, diffusion, and bioturbation, respectively, are directly visible in the time series, while for the tree model, such an effect is neither expected nor

[Figure]

**Figure S5.** Normalised histograms, autocorrelation functions (acf) and estimated power spectral densities (psd) of the different input and proxy system model output time series for GWN, AR(1), ROS, LOR, and LMR (top to bottom). The input variables are denoted in grey, the tree model output in green, the lake model output in blue, the speleothem model output in red, and the ice model output in magenta.

visible. Also, it should be noted that the tree ring model with the parameters as specified in Table 1 seems to primarily respond to temperature rather than to precipitation, meaning that the limiting factor for tree growth in eastern Canada is temperature, which is ecologically reasonable and also be related to the fact that precipitation is only indirectly taken into account in the used model for tree ring width.

For further evaluation, we standardise all time series to zero mean and unit variance and examine some properties of the different input and output series. The left panels of Fig. S5 show the normalised histograms of the input and output variables. To quantify differences in the histograms, we consider the skewness of the distributions of the different time series (see Table S1). We observe that the resulting time series from the tree model all show significant deviations in skewness from the input data, which is probably due to the thresholding of the growth response functions for temperature and soil moisture. For the other proxy models, we observe a significant influence of the model on the skewness of the output for those inputs that are not

20   stochastic. This might either be related to the smaller confidence bounds due to the different surrogate generation, or hint to a
different reaction of the models to different distributions of the input data.

Finally, we have a look at the autocorrelation functions and the power spectral densities of the different input and proxy
model output time series. The middle and right panels of Fig. S5 show the resulting autocorrelation functions and estimates
of the power spectral density obtained using the Welch method (Welch, 1967). As before, we observe that the autocorrelation

25   function and power spectral density of the tree ring model output closely follow the respective temperature input. The good
agreement between the temperature input and the tree ring width model output can be partially attributed to the model not
taking into account juvenile growth of the trees. The speleothem model output shows the expected loss of power in the higher
frequencies. Similarly, a loss of power in the higher frequencies can be observed in the ice core model output but not as
pronounced as for the speleothem model output. The same holds for the lake sediment model output except for the Rössler and

30   Lorenz scenarios, where the lake model output shows more power in the higher frequencies than the corresponding input. This
is most likely related to the different time-scales of variability present in those time series in comparison with the others and
the particular choice of mixed layer thickness in the lake sediment model, mxl = 4.

**Skewness of time series**

Table S1 gives the skewness of the distributions of the different time series where those values showing a significant deviation

35   from the skewness of the corresponding input are marked in bold. To test whether the skewness values of the model output time
series differ significantly from that of the respective input, we create $N_{sk} = 10,000$ surrogate data sets for each of the input time
series. For GWN and the AR(1) process, this is done by creating different realisations of the corresponding process according
to the descriptions in Sec. 4, while for the other time series, the surrogates are created by adding white noise with signal-to-
noise ratios of 25 for the model systems and 100 for the last millennium reanalysis data. Then, for each surrogate realisation,

40   we calculate the skewness of the distribution and take the 0.5th and 99.5th percentile as confidence bounds resulting in the 99%
confidence intervals of $[-0.20, 0.20]$ for GWN, $[-0.29, 0.29]$ for the AR(1) process, $[0.20, 0.23]$ for the non-stationary Rössler
system, $[-0.03, 0.00]$ for the non-stationary Lorenz system, $[0.32, 0.34]$ for the last millennium reanalysis temperature, and
$[0.00, 0.02]$ for the last millennium reanalysis precipitation. As precipitation is proportional to the negative temperature and
isotopes, the corresponding confidence interval for the Rössler system is $[-0.23, -0.20]$ and for the Lorenz system $[0.00, 0.03]$.

45

**Table S1.** Skewness of the different input and proxy system model output time series with significant deviations of the model output time series from the corresponding input marked in bold.

| input/output | GWN | AR(1) | ROS | LOR | LMR |
|---|---|---|---|---|---|
| temperature | 0.00 | 0.07 | 0.21 | −0.02 | 0.33 |
| precipitation | 0.12 | −0.03 | −0.20 | 0.01 | 0.01 |
| isotopes | −0.06 | 0.12 | 0.22 | −0.02 | – |
| trw | **0.34** | **0.35** | **0.40** | **0.26** | **0.63** |
| lak | 0.11 | 0.23 | **−0.05** | **0.01** | **0.71** |
| spt | −0.04 | 0.29 | **−0.27** | **0.26** | – |
| ice | −0.01 | 0.15 | **0.15** | −0.05 | – |